# POLYMORPHIC: PLUG-AND-PLAY VISUAL TOKEN COMPRESSION FOR SCALABLE VLMS

## ABSTRACT

Recent advances in vision language models (VLMs) have enabled strong reasoning and generalization capabilities, but they remain computationally expensive, primarily due to the quadratic complexity of Transformer self-attention and the large number of visual tokens produced by high-resolution inputs. To address this limitation, we propose a flexible plug-and-play framework for visual token pruning that can be seamlessly integrated into existing VLMs without requiring additional training or model modification. Our approach employs a two-stage strategy. In the first stage, representation-level token merging is performed based on spatial information density, which removes redundant visual features. In the second stage, tokens with low cross-modal relevance are adaptively pruned during language model prefilling, allowing the computation to focus on the most informative regions. This design substantially reduces the visual token budget, leading to improvements in both inference speed and memory efficiency while maintaining strong task performance. Extensive experiments on widely used benchmarks demonstrate that our method consistently achieves superior efficiency and accuracy trade-offs, highlighting its potential for practical deployment of high-resolution VLMs in real-world applications.

## 1 INTRODUCTION

Recent progress in large language models (LLMs) (Ding et al., 2022; Qin et al., 2023; Zhu et al., 2023; Li et al., 2023a) has catalyzed rapid advances in vision language models (VLMs) (Bai et al., 2025; Team et al., 2025; Zhu et al., 2025), which are now pervasive across a broad spectrum of downstream applications, ranging from visual question answering to multimodal dialogue. A standard VLM architecture (Liu et al., 2024a; Lin et al., 2023; Zhang et al., 2025) is typically composed of three principal modules: a visual encoder (e.g., CLIP or SigLIP) responsible for extracting visual features from images or videos, a lightweight projector that maps these features into the semantic embedding space of the language model, and a large language model that serves as the dominant reasoning engine, generating responses in an auto regressive manner.

Although recent VLMs exhibit remarkable reasoning and generalization capabilities, their computational footprint remains prohibitively large (Liu et al., 2024c; Shang et al., 2024b; Huang et al., 2024; Tong et al., 2025; Li et al., 2025a). The primary bottleneck arises from the quadratic complexity of Transformer self attention, which is further exacerbated by the disproportionate number of visual tokens. High resolution images can easily produce thousands of tokens, whereas text inputs typically span only a few dozen (Zhang et al., 2024a; Chen et al., 2024a; Lin et al., 2025b; Huang et al., 2024; Sun et al.). This imbalance introduces both inefficiency and memory strain, ultimately hindering practical deployment in latency sensitive or resource constrained scenarios, such as edge devices and real time interactive systems.

To alleviate this challenge, we introduce a novel plug and play token pruning framework tailored for VLMs. Our approach integrates two complementary stages. First, we exploit the spatial distribution of features from the visual encoder to identify and merge tokens in regions characterized by low information density, thereby reducing redundancy at the representation level. Second, during LLM decoding, we adaptively prune visual tokens with weak query relevance, as measured by cross modal attention dynamics. This two stage mechanism substantially reduces the visual token budget, yielding notable improvements in both computational and memory efficiency. Importantly,

our method operates entirely without fine tuning or additional training, making it directly applicable to existing off the shelf VLMs. Extensive experiments on widely adopted benchmarks demonstrate that our approach consistently delivers superior efficiency versus performance trade offs, achieving significant inference acceleration while preserving competitive accuracy across diverse tasks.

Our main contriubutions can be summarized as follows:

- We introduce a new plug and play framework for visual token pruning in vision language models. The method is model agnostic and requires no additional fine tuning, which allows immediate application to existing systems.

- We design a two stage pruning strategy that first merges tokens in spatial regions with low information density and then prunes tokens with weak text guided relevance during large language model decoding.

- We conduct extensive experiments on widely used benchmarks and demonstrate that our method achieves a favorable balance between efficiency and performance, outperforming prior approaches.

## 2  RELATED WORK

### 2.1  VISION-LANGUAGE MODELS

LLaVA (Liu et al., 2023) typically encodes a $336 \times 336$ image into 576 tokens, while LLaVA-Next (Liu et al., 2024b) adopts a hybrid strategy: it first resizes the image into a $336 \times 336$ global view encoded as 576 tokens, and then selects the optimal resolution for the raw image, resizing and partitioning it into up to four quadrants, which together yield up to 2880 tokens (Chen et al., 2024b; Lin et al., 2025a; Shang et al., 2024a; Tong et al., 2025). A similar trend can be observed in the Qwen family of models, where the number of visual tokens steadily increases from Qwen2-VL (Wang et al., 2024) to Qwen2.5-VL in order to enhance visual understanding. However, since Transformer self-attention scales quadratically with sequence length, the computational cost increases significantly as more visual tokens are introduced.

### 2.2  VISUAL TOKEN PRUNING AND COMPRESSION

Recent studies explore pruning and compression to alleviate the high cost of visual token processing (Ye et al., 2025; Lin et al., 2025a; Li et al., 2025b; Cai et al., 2024). FastV (Chen et al., 2024a) prunes tokens at a fixed LLM layer based on attention scores, while VTW (Lin et al., 2025a) assumes early absorption of visual information and drops all tokens after a certain layer. Encoder-level methods, such as VisionZip (Yang et al., 2025), reveal that CLIP and SigLIP exhibit highly concentrated attention, enabling pruning guided by [CLS]-based distributions. In contrast, SparseVLM (Zhang et al., 2024b) introduces text-guided pruning that progressively removes tokens with low cross-modal attention, achieving competitive accuracy but requiring all tokens to first pass through the LLM, which limits efficiency for high-resolution inputs.

## 3  METHOD

### 3.1  OVERVIEW

Our approach consists of a two-stage framework to efficiently handle visual tokens in vision-language models. In the first stage, sparse or redundant visual tokens are identified and merged using an adaptive clustering algorithm that groups tokens based on feature similarity and local density. This produces a compressed representation that preserves critical visual information while substantially reducing sequence length. In the second stage, the compressed tokens are dynamically pruned during the LLM's prefill process. We select semantically meaningful text tokens as raters to evaluate cross-modal alignment, guiding the pruning of less informative visual tokens. This two-stage mechanism effectively reduces computational overhead while maintaining high-quality visual representations for downstream reasoning tasks. Figure 1 provides an overview of our proposed method.

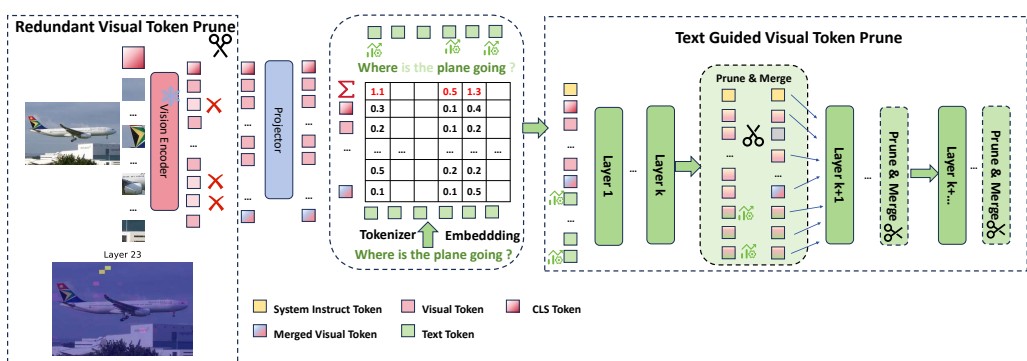

Figure 1: Overview and Core Design of the Proposed Framework.

## 3.2 GLOBAL–LOCAL CONSOLIDATION OF SPARSE VISUAL EMBEDDINGS

Some studies (Liu et al., 2024c; Shang et al., 2024b; Huang et al., 2024; Tong et al., 2025; Li et al., 2025a) have found that attention scores are crucial for evaluating the importance of tokens. We leverage the attention scores produced by the model to quantitatively assess the importance of each visual token. Specifically, tokens receiving higher attention weights are considered more informative and influential for downstream reasoning, whereas tokens with lower attention contributions are deemed less critical and may be candidates for pruning or consolidation. This mechanism enables a fine-grained, data-driven prioritization of tokens, allowing the model to focus computational resources on the most salient regions of the input while maintaining representational fidelity. Based on this observation, we propose a token consolidation strategy that merges tokens carrying sparse or secondary information. The key intuition is that semantically similar tokens can be aggregated with minimal information loss, thereby preserving essential content while reducing the token budget. To this end, we introduce an adaptive clustering algorithm that groups tokens according to both feature similarity and local density, and subsequently replaces each cluster with a representative token. Formally, let the output of the visual encoder be denoted as $\mathbf{X} \in \mathbb{R}^{N \times C}$, where $N$ is the number of tokens and $C$ is the feature dimension. The goal is to compress the sequence length from $N$ to $K$, yielding a compact representation $\mathbf{X}' \in \mathbb{R}^{K \times C}$ that maintains both semantic fidelity and computational efficiency. Pairwise dissimilarities between tokens are quantified as follows:

$$\mathbf{D}_{i,j} = \frac{1}{\sqrt{C}} \|\mathbf{X}_{i,:} - \mathbf{X}_{j,:}\|_2 \tag{1}$$

or equivalently in matrix form

$$\mathbf{D} = \sqrt{\frac{1}{C}} \left( \operatorname{diag}(\mathbf{X}\mathbf{X}^\top)\mathbf{1}^\top + \mathbf{1}\operatorname{diag}(\mathbf{X}\mathbf{X}^\top)^\top - 2\mathbf{X}\mathbf{X}^\top \right)^{1/2} \tag{2}$$

Local density for token $i$ is estimated using its $K_{\mathrm{nn}}$ nearest neighbors

$$\boldsymbol{\rho}_i = \exp\left( -\frac{1}{K_{\mathrm{nn}}} \sum_{k=1}^{K_{\mathrm{nn}}} \mathbf{D}_{i,n_k(i)}^2 \right) \quad \text{or equivalently as} \quad \boldsymbol{\rho} = \exp\left( -\frac{\mathbf{D}_{\mathrm{nn}}^2}{K_{\mathrm{nn}}} \right) \tag{3}$$

where $\mathbf{D}_{\mathrm{nn}}$ is the $N \times K_{\mathrm{nn}}$ nearest-neighbor distance submatrix. This density can also be formulated as a kernel density estimate over all tokens

$$\boldsymbol{\rho}_i \propto \sum_{j=1}^{N} \exp\left( -\frac{\mathbf{D}_{i,j}^2}{\sigma^2} \right) \tag{4}$$

with bandwidth $\sigma$ controlling the smoothness. To balance local density and global coverage, we define a prominence score for token $i$

$$\mathbf{S}_i = \boldsymbol{\rho}_i \cdot \delta_i \qquad \delta_i = \min_{j:\, \boldsymbol{\rho}_j > \boldsymbol{\rho}_i} \mathbf{D}_{i,j} \tag{5}$$

Selecting the top-$K$ scores yields cluster centers $\mathbf{I}_c$. Each token is assigned to its nearest center, inducing clusters $\mathcal{C}_k$

$$\mathbf{A}_i = \arg\min_{j \in \mathbf{I}_c} \mathbf{D}_{i,j} \quad \mathcal{C}_k = \{i \mid \mathbf{A}_i = k\} \tag{6}$$

Feature aggregation within each cluster is expressed as

$$\mathbf{X}'_{k,:} = \frac{1}{|\mathcal{C}_k|} \sum_{i \in \mathcal{C}_k} \mathbf{X}_{i,:} \quad \text{or equivalently in matrix form:} \quad \mathbf{X}' = \mathbf{M}_{\mathrm{agg}}\mathbf{X} \tag{7}$$

where $(\mathbf{M}_{\mathrm{agg}})_{k,i} = \frac{1}{|\mathcal{C}_k|}$ if $i \in \mathcal{C}_k$ and 0 otherwise. We can further formulate this consolidation as an optimization problem that minimizes intra-cluster distances while preserving density

$$\mathbf{X}' = \arg\min_{\mathbf{Y} \in \mathbb{R}^{K \times C}} \sum_{k=1}^{K} \sum_{i \in \mathcal{C}_k} \boldsymbol{\rho}_i \|\mathbf{X}_{i,:} - \mathbf{Y}_{k,:}\|_2^2 \tag{8}$$

Through this approach, we effectively perform token merging, whereby the merged tokens are combined with the high attention score retained tokens and fed into the LLM. This strategy substantially enhances inference efficiency by reducing sequence length and increasing information density.

### 3.3 ADAPTIVE CROSS-MODAL VISUAL TOKEN PRUNING

Even after cluster consolidation, sequences can remain long, imposing substantial computation. To address this, we propose a dynamic pruning mechanism guided by cross-modal relevance. Not all text tokens provide meaningful guidance; punctuation or common function words may introduce noise. We therefore select a subset of text tokens with strong alignment as *raters* Let $\mathbf{V} \in \mathbb{R}^{N_v \times d}$ be visual token embeddings and $\mathbf{T} \in \mathbb{R}^{N_t \times d}$ be textual token embeddings. The cross-modal similarity is computed via scaled dot-product

$$\mathbf{M} = \mathrm{softmax}\left(\frac{\mathbf{V}\mathbf{T}^\top}{\sqrt{d}}\right) \tag{9}$$

where the softmax is applied along the text-token dimension. The mean alignment score for each text token is

$$\mathbf{s}_j = \frac{1}{N_v} \sum_{i=1}^{N_v} \mathbf{M}_{i,j} \qquad \bar{\mathbf{s}} = \frac{1}{N_t} \sum_{j=1}^{N_t} \mathbf{s}_j \tag{10}$$

and raters are selected as $\mathcal{R} = \{j \mid \mathbf{s}_j > \bar{\mathbf{s}}\}$. The importance of each visual token is defined as its average alignment with any rater

$$\alpha_i = \mathrm{Avg}_{j \in \mathcal{R}} \mathbf{M}_{i,j} \tag{11}$$

Pruning is applied according to a threshold $\tau$, $\tau$ is the token score quantile computed according to the pruning rate, represented as follows:

$$\mathcal{V}_{\mathrm{kept}} = \{i \mid \alpha_i \geq \tau\} \quad \mathcal{V}_{\mathrm{pruned}} = \{i \mid \alpha_i < \tau\} \tag{12}$$

For pruned tokens, we apply the merging strategy described in Section 3.2 again to further enhance information retention.

## 4 EXPERIMENTS

### 4.1 EXPERIMENTAL SETUP

**Datasets** For image-based multimodal evaluation, we conduct experiments on six widely adopted benchmarks, including TextVQA (Singh et al., 2019), ScienceQA (SQA) (Saikh et al., 2022), POPE (Li et al., 2023b), MME (Fu et al., 2023), GQA (Hudson & Manning, 2019), and VizWiz (Bigham et al., 2010). These datasets collectively cover diverse capabilities such as visual question answering, knowledge reasoning, and robustness to adversarial prompts, thereby providing a comprehensive evaluation of model performance under different multimodal understanding scenarios.

**Models** To demonstrate the effectiveness and versatility of our framework, we evaluate it on three representative VLMs: LLaVA-v1.5 (7/13B) (Liu et al., 2023), LLaVA-NeXT (7/13B) (Liu et al., 2024b),, and Qwen2-VL (7B) (Wang et al., 2024). For models such as LLaVA-v1.5 and LLaVA-NEXT, whose visual encoders incorporate a [CLS] token, we exploit the attention distribution of the [CLS] token to guide visual token pruning. In contrast, for Qwen2-VL, which does not employ a [CLS] token in its visual encoder, we instead leverage the averaged attention scores across all visual tokens. This unified design ensures that our approach remains broadly applicable across diverse VLM architectures and visual encoding paradigms.

**Implementation Deatils** We conduct all experiments on machines equipped with four NVIDIA A800 GPUs, implemented using the PyTorch framework. In the first stage, we prune visual tokens according to their information distribution. For LLaVA and LLaVA-NeXT, the attention score of the [CLS] token is used as a global indicator of semantic relevance, whereas Qwen2-VL, which does not employ a [CLS] token, relies on the average attention score across all visual tokens. During Pruning of Low-Density Visual Tokens, we retain approximately $K = 2$ times the target budget to preserve potentially informative tokens. Tokens with sparse content are merged into compact clusters, yielding a retained-to-merged ratio of about $5.4 : 1$. In the second stage, the compressed tokens are further pruned within the LLM during the prefill process. Pruning is applied at fixed layers to maintain a consistent average number of visual tokens across the network. For LLaVA-v1.5 (7B with 32 layers and 13B with 40 layers), pruning is performed at layers 3, 7, and 16. For Qwen2-VL-7B-Instruct, pruning occurs at layers 3, 7, and 13.

Table 1: Performance comparison between baselines and Ours on LLaVA-v1.5-7B across budgets.

| Token Budget | Method | Vizwiz | SQA | GQA | TextVQA | MME | POPE (F1) | Mean % |
|---|---|---|---|---|---|---|---|---|
| 576 | Vanilla | 54.04 | 69.41 | 61.92 | 46.00 | 1864.84 | 85.85 | 100.00% |
| 192 | FastV | 56.87 | 55.23 | 53.94 | 41.45 | 1556.45 | 79.33 | 89.65% |
| | SparseVLMs | 54.04 | 68.96 | 59.33 | 44.68 | 1781.43 | 85.41 | **97.89%** |
| | VisionZip | 53.98 | 68.62 | 59.22 | 44.58 | 1769.02 | 85.29 | 97.58% |
| | Ours | 52.30 | 68.32 | 59.95 | 45.18 | 1772.17 | 85.47 | 97.47% |
| 128 | FastV | 56.26 | 55.43 | 51.26 | 37.76 | 1490.22 | 72.52 | 85.54% |
| | SparseVLMs | 53.11 | 68.86 | 58.36 | 42.43 | 1743.33 | 84.70 | 96.02% |
| | VisionZip | 54.05 | 68.67 | 57.62 | 43.79 | 1768.73 | 82.89 | 96.44% |
| | Ours | 52.25 | 68.27 | 59.00 | 44.56 | 1780.38 | 85.02 | **96.95%** |
| 64 | FastV | 51.51 | 53.30 | 38.03 | 15.61 | 1045.11 | 17.70 | 57.35% |
| | SparseVLMs | 52.75 | 68.96 | 52.46 | 32.10 | 1601.96 | 76.17 | 87.68% |
| | VisionZip | 54.68 | 68.96 | 55.16 | 41.95 | 1716.93 | 76.97 | 93.76% |
| | Ours | 52.72 | 69.46 | 56.20 | 42.60 | 1734.67 | 81.79 | **94.88%** |

## 4.2 MAIN RESULTS

**Results on LLaVA-v1.5** As shown in Tables 1 and 2, our method demonstrates strong robustness across token budgets on both LLaVA-v1.5-7B and LLaVA-v1.5-13B. On LLaVA-v1.5-7B, Sparse-VLM achieves the highest relative performance at 192 and 128 tokens (97.89% and 96.02%), while our method remains competitive (97.47% and 96.95%). FastV degrades significantly (89.65% and 85.54%), whereas VisionZip is stable (97.58% and 96.44%). At 64 tokens, SparseVLM drops to 87.68%, VisionZip reaches 93.76%, and our method attains 94.88%, the highest robustness. A similar trend appears on LLaVA-v1.5-13B: SparseVLM leads at 192/128 tokens (98.02% and 97.28%), our method remains competitive (97.60% and 97.09%), FastV lags (96.24% and 91.70%), and VisionZip is comparable (97.25% and 96.10%). Under 64 tokens, FastV (73.52%) and SparseVLM (87.84%) degrade sharply, while VisionZip (93.56%) and our method (95.25%) maintain strong performance, with ours being the most robust. In summary, SparseVLM excels under moderate pruning, but our method consistently outperforms all baselines under aggressive compression. FastV suffers severe drops, and VisionZip, though stable, lags behind our approach at extreme token reduction.

**Results on Qwen2-VL** Unlike LLaVA-based models, Qwen2-VL employs a dynamic resolution encoder without a [CLS] token. By leveraging the average attention score across all visual tokens, our method naturally adapts to this architecture. As shown in Table 3, our approach con-

Table 2: Performance comparison between baselines and ours on LLaVA-v1.5-13B across budgets.

| Token Budget | Method | Vizwiz | SQA | GQA | TextVQA | MME | POPE (F1) | Mean % |
|---|---|---|---|---|---|---|---|---|
| 576 | Vanilla | 56.18 | 72.78 | 63.29 | 48.77 | 1823.95 | 85.96 | 100.00% |
| 192 | FastV | 60.97 | 68.52 | 59.26 | 45.52 | 1679.06 | 82.27 | 96.24% |
| | SparseVLMs | 55.08 | 73.18 | 60.06 | 47.62 | 1793.92 | 84.82 | **98.02%** |
| | VisionZip | 54.99 | 73.53 | 59.12 | 46.97 | 1749.59 | 85.04 | 97.25% |
| | Ours | 54.86 | 73.57 | 59.46 | 47.19 | 1759.19 | 85.68 | 97.60% |
| 128 | FastV | 60.11 | 68.47 | 56.65 | 40.42 | 1614.04 | 75.88 | 91.70% |
| | SparseVLMs | 54.67 | 73.87 | 58.59 | 45.79 | 1840.54 | 83.81 | **97.28%** |
| | VisionZip | 54.64 | 74.12 | 57.76 | 46.44 | 1739.74 | 82.22 | 96.10% |
| | Ours | 54.65 | 73.48 | 59.10 | 46.54 | 1773.50 | 84.47 | 97.09% |
| 64 | FastV | 54.45 | 67.28 | 46.57 | 23.70 | 1326.20 | 48.89 | 73.52% |
| | SparseVLMs | 54.39 | 72.53 | 54.40 | 34.96 | 1620.02 | 72.31 | 87.84% |
| | VisionZip | 55.95 | 74.22 | 56.03 | 44.10 | 1686.40 | 76.00 | 93.56% |
| | Ours | 54.96 | 73.48 | 57.55 | 45.14 | 1748.98 | 80.22 | **95.25%** |

sistently surpasses VisionZip across all token budgets, with particularly large improvements under moderate pruning where VisionZip suffers substantial degradation. Specifically, our method achieves 95.07% and 92.91% mean retention at 192 and 128 tokens, respectively, compared to only 83.86% for VisionZip at both settings. Even under the aggressive 64-token budget, our method attains 85.63%, outperforming VisionZip (83.77%). The gains are especially pronounced on tasks requiring fine-grained reasoning and visual-text alignment, where VisionZip often collapses. These results demonstrate that our method not only preserves robustness under aggressive pruning but also scales effectively to heterogeneous VLM architectures beyond the LLaVA family.

Table 3: Performance comparison between baselines and ours on Qwen2-VL-7B-Instruct across budgets.

| Token Budget | Method | Vizwiz | SQA | GQA | TextVQA | MME | POPE (F1) | Mean % |
|---|---|---|---|---|---|---|---|---|
| Full | Vanilla | 68.84 | 84.98 | 62.39 | 81.27 | 2295.23 | 87.76 | 100.00 |
| 192 | VisionZip | 61.83 | 77.00 | 57.39 | 37.65 | 1963.69 | 86.76 | 83.86 |
| | Ours | 66.87 | 81.66 | 61.09 | 66.59 | 2242.49 | 87.43 | **95.07** |
| 128 | VisionZip | 61.82 | 77.19 | 57.49 | 37.75 | 1955.99 | 86.65 | 83.86 |
| | Ours | 65.51 | 82.05 | 60.13 | 58.57 | 2255.20 | 86.92 | **92.91** |
| 64 | VisionZip | 61.83 | 77.00 | 57.39 | 37.61 | 1954.01 | 86.70 | 83.77 |
| | Ours | 61.89 | 78.24 | 56.82 | 41.17 | 2113.52 | 86.00 | **85.63** |

## 4.3 EFFICIENCY

As shown in Table 4, the majority of efficiency gains arise in the prefill stage, where the quadratic complexity of self-attention with respect to sequence length renders visual token pruning particularly effective. On LLaVA-v1.5-7B, VisionZip achieves the fastest prefill time (24.10 ms) with a 2.28× speedup, while our proposed method also reduces prefill latency to 32.44 ms, corresponding to a 1.70× improvement. In contrast, the decode stage exhibits only minor differences across methods. On the LLaVA-NEXT-7B, the benefits of pruning are further amplified due to the substantially higher initial token budget: VisionZip reaches an 8.80× overall speedup, while ours achieves a 6.70× acceleration, confirming the effectiveness of our two-stage pruning strategy in mitigating prefill overhead while maintaining stable decoding efficiency.

As shown in Table 4, the majority of efficiency gains arise in the prefill stage, where the quadratic complexity of self-attention with respect to sequence length makes visual token pruning particularly effective. On LLaVA-v1.5-7B, VisionZip achieves the fastest prefill time (24.10 ms) with a 2.28× speedup, while our method also reduces prefill latency to 32.44 ms, corresponding to a 1.70× improvement. For LLaVA-NEXT-7B, the benefits of pruning are further amplified due to its substantially larger initial token budget: VisionZip reaches an 8.80× overall speedup, while our method

achieves a 6.70× acceleration, confirming the effectiveness of our strategy in mitigating prefill overhead while maintaining stable decoding efficiency. The superior acceleration of VisionZip primarily stems from its design, where all visual tokens are pruned down to the target budget before being fed into the LLM, thereby directly shortening the sequence length in prefill computation. In contrast, our two-stage approach retains richer visual information during the initial encoding and adaptively prunes tokens in the prefill stage based on text–visual relevance, resulting in slightly lower raw speedup but offering more favorable accuracy–efficiency trade-offs.

Table 4: Prefill and Decode Time Comparison (ms) and Speedup for Different Methods.

| Model | Method | Budget | Time (ms) | | Speedup |
|---|---|---|---|---|---|
| | | | Prefill | Decode | |
| LLaVA-v1.5-7B | Vanilla | 576 | 55.05 | 21.20 | 1.00× |
| | SparseVLM | 64 | 36.40 | 26.69 | 1.51× |
| | VisionZip | 64 | 24.10 | 21.31 | **2.28×** |
| | Ours | 64 | 32.44 | 21.76 | 1.70× |
| LLaVA-NEXT-7B | Vanilla | 2880 | 248.02 | 22.49 | 1.00× |
| | VisionZip | 160 | 28.19 | 21.31 | **8.80×** |
| | Ours | 160 | 37.02 | 22.63 | 6.70× |

## 4.4 PERFORMANCE UNDER DIFFERENT PRUNING BUDGETS

Figures 2a, 2b, 2c, and 2d report the results of different methods on GQA, MME, POPE, and TextVQA under varying visual token budgets. Our approach (denoted as Ours) consistently achieves strong performance across all tasks and budget levels, demonstrating robustness under aggressive pruning while remaining competitive when more tokens are preserved. In contrast, FastV suffers severe degradation, particularly under low-token budgets, indicating that pruning solely based on CLS-token attention is unreliable as it often discards tokens critical for downstream reasoning. SparseVLM performs competitively under moderate and large budgets, but its performance drops sharply when tokens are extremely limited, suggesting that the absence of explicit filtering for visually uninformative tokens leads to misleading pruning signals. VisionZip exhibits stronger resilience under extreme pruning, sometimes surpassing both FastV and SparseVLM at very low budgets. However, its advantage diminishes as the budget increases, where it falls behind both SparseVLM and our method. Overall, our method provides the most stable efficiency–accuracy trade-off. At high budgets (192 and 256 tokens), Ours matches or surpasses SparseVLM and VisionZip across all benchmarks. At low budgets (32 and 64 tokens), it achieves substantially higher accuracy than FastV and remains competitive with VisionZip, while avoiding the sharp degradation observed in SparseVLM. These results highlight the effectiveness of our two-stage framework redundancy reduction and cross-modal guided pruning—which together enable robust performance across a wide range of pruning scenarios.

## 4.5 HYPERPARAMETER ANALYSIS

In our framework, the hyperparameter $K$ controls the degree of redundancy retained after the first-stage pruning based on visual information distribution. Specifically, we initially preserve $K$ times the target visual token budget before applying the second-stage text-guided pruning, ensuring that the average number of visual tokens across LLM layers matches the final budget. A smaller $K$ implies that more pruning is performed purely at the representation level, which risks discarding tokens that may contain task-relevant fine-grained details. When $K = 1$, the framework essentially relies solely on text-guided pruning. Conversely, a larger $K$ retains more tokens after the first stage, shifting the burden of pruning to the second stage, where cross-modal attention dynamics determine their utility with respect to the input query. Thus, $K$ essentially balances spatial redundancy removal against text-guided adaptivity.

Table 5 reports results on LLaVA-v1.5-7B across four benchmarks (GQA, TextVQA, MME, and POPE). At the 128-token budget, $K = 1$ performs the worst (93.95%), indicating that exclusive reliance on text-guided pruning is suboptimal. Increasing $K$ improves performance, with $K = 2$

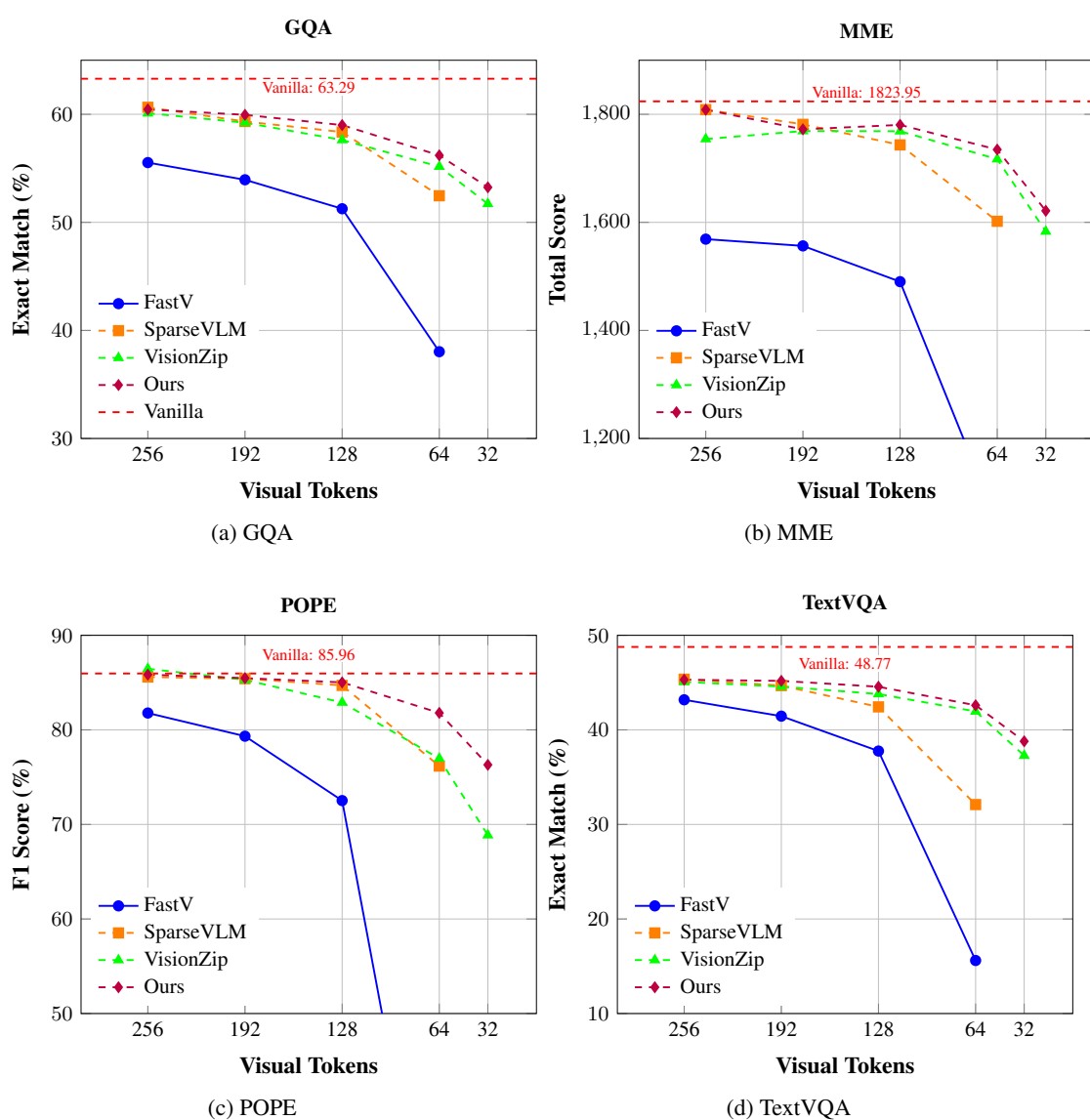

Figure 2: Performance comparison across different budgets and methods on four benchmarks.

achieving the best mean score (96.66%). Beyond this point, larger $K$ values (e.g., 2.5 or 3) slightly degrade results, likely due to excessive reliance on text-guided pruning, which dilutes spatial redundancy removal and reduces robustness on visually demanding tasks. Under the 64-token budget, the best result is observed at $K = 2.5$ (93.12%), but the overall trend is consistent: moderate redundancy retention in the first stage is critical, while both overly small ($K = 1$) and overly large ($K = 3$) values lead to performance drops. These findings suggest that $K = 2$ offers a robust default choice across different budgets, striking a favorable balance between efficiency and accuracy.

## 4.6 ABLATION STUDY

We conduct ablations on LLaVA-v1.5-7B across GQA, TextVQA, MME, and POPE to assess the contribution of each module. Two variants are considered: ablation$_1$, which removes Cross-Modal Guided Token Pruning, and ablation$_2$, which removes Pruning of Low-Density Visual Tokens. As shown in Table 6, both ablations consistently underperform the full model. Our method achieves the highest relative scores across all token budgets, retaining 97.41% of baseline accuracy at 192 tokens and maintaining robustness under aggressive pruning (92.92% at 64 tokens). These results

Table 5: Performance comparison of different K values on LLaVA-v1.5-7B under fixed token budgets.

| Budget | K | GQA | TextVQA | MME | POPE (F1) | Mean % |
|---|---|---|---|---|---|---|
| 576 | - | 61.92 | 46.00 | 1864.84 | 85.85 | 100.00% |
| 128 | 1 | 58.33 | 42.40 | 1719.43 | 83.47 | 93.95% |
| | 1.5 | 58.79 | 44.35 | 1764.12 | 85.33 | 96.34% |
| | 2 | 59.00 | 44.56 | 1780.38 | 85.02 | **96.66%** |
| | 2.5 | 59.17 | 44.09 | 1762.54 | 84.88 | 96.20% |
| | 3 | 58.53 | 42.62 | 1772.28 | 84.37 | 95.12% |
| 64 | 1.5 | 56.08 | 42.45 | 1697.62 | 81.16 | 92.10% |
| | 2 | 56.20 | 42.60 | 1734.67 | 81.79 | 92.92% |
| | 2.5 | 56.54 | 41.69 | 1755.58 | 82.75 | **93.12%** |
| | 3 | 55.75 | 38.87 | 1742.08 | 83.02 | 91.16% |

demonstrate that the two modules are complementary, and their integration is crucial for achieving superior efficiency–accuracy trade-off.

Table 6: Ablation study on LLaVA-v1.5-7B. Removing either Cross-Modal Guided Token Pruning or Pruning degrades performance, confirming that both modules are complementary.

| Token Budget | Method | GQA | TextVQA | MME | POPE (F1) | Mean % |
|---|---|---|---|---|---|---|
| 576 | Vanilla | 61.92 | 46.00 | 1864.84 | 85.85 | 100.00% |
| 192 | ablation$_1$ | 59.35 | 44.63 | 1773.24 | 85.76 | 96.96% |
| | ablation$_2$ | 59.16 | 44.57 | 1774.68 | 85.11 | 96.68% |
| | Ours | 59.95 | 45.18 | 1772.17 | 85.47 | **97.41%** |
| 128 | ablation$_1$ | 57.70 | 43.89 | 1758.08 | 84.47 | 95.32% |
| | ablation$_2$ | 58.33 | 42.40 | 1719.43 | 83.47 | 93.95% |
| | Ours | 59.00 | 44.56 | 1780.38 | 85.02 | **96.66%** |
| 64 | ablation$_1$ | 55.18 | 41.83 | 1675.53 | 77.38 | 90.00% |
| | Ours | 56.20 | 42.60 | 1734.67 | 81.79 | **92.92%** |

## 5 CONCLUSION

We introduce a versatile plug-and-play framework for visual token pruning that can be directly incorporated into existing vision–language models (VLMs) without additional training or structural modification. The framework adopts a two-stage procedure. First, representation-level token merging guided by spatial information density removes redundant visual features. Second, during language decoding, tokens with low cross-modal relevance are adaptively pruned, enabling computation to concentrate on the most informative regions. This design markedly reduces the visual token budget, thereby improving inference speed and memory efficiency while preserving strong task performance. Extensive experiments on standard benchmarks verify that our method consistently attains superior efficiency–accuracy trade-offs, underscoring its potential for practical deployment of high-resolution VLMs in real-world scenarios.

ETHICS STATEMENT

This work adheres to the ICLR Code of Ethics. All datasets utilized in this paper are publicly available and widely adopted within the research community, and we strictly follow their respective licenses and intended usage.

REPRODUCIBILITY STATEMENT

We strive to ensure the reproducibility of our results. Full details are provided in the main paper and the appendix. Our implementation is built on PyTorch and standard open-source libraries.

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

## A  RESULTS ON LLAVA-NEXT

SparseVLM feeds all visual tokens to the LLM without filtering based on their information distribution. This design becomes less effective for LLaVA-NeXT, which supports high-resolution inputs and thus generates a substantially larger number of visual tokens. As a result, SparseVLM struggles to sustain performance under low-token budgets, and we omit its results in this setting. FastV also faces difficulties with LLaVA-NeXT due to its reliance on token merging, which leads to degraded performance under high-resolution conditions. As shown in Table 7 and 8, we report results under token budgets of 640, 320, and 160. On both LLaVA-NeXT-7B and LLaVA-NeXT-13B, our method consistently outperforms VisionZip across all budgets, highlighting its robustness and adaptability to models with high-resolution visual encoders.

Table 7: Performance comparison between baselines and ours on LLaVA-NEXT-7B across budgets.

| Budget | Method | Vizwiz | SQA | GQA | TextVQA | MME | POPE (F1) | Mean % |
|---|---|---|---|---|---|---|---|---|
| 2880 | Vanilla | 60.70 | 70.25 | 64.26 | 64.80 | 1849.30 | 86.43 | 100.00% |
| 640 | FastV | 58.86 | 56.37 | 57.51 | 55.80 | 1685.20 | 83.51 | 90.09% |
| | VisionZip | 60.80 | 67.67 | 61.35 | 61.31 | 1800.80 | 86.12 | 97.26% |
| | Ours | 59.84 | 68.57 | 62.38 | 61.51 | 1815.64 | 86.76 | **97.79%** |
| 320 | FastV | 55.41 | 55.68 | 49.56 | 37.29 | 1405.70 | 64.75 | 76.03% |
| | VisionZip | 59.80 | 67.63 | 59.06 | 58.33 | 1734.48 | 82.44 | 94.32% |
| | Ours | 60.01 | 67.87 | 60.78 | 59.16 | 1809.23 | 83.72 | **96.01%** |
| 160 | VisionZip | 60.02 | 68.02 | 55.41 | 52.45 | 1622.37 | 75.11 | 89.58% |
| | Ours | 59.34 | 67.77 | 58.17 | 55.49 | 1687.10 | 78.83 | **92.14%** |

Table 8: Performance comparison between baselines and ours on LLaVA-NEXT-13B across budgets.

| Budget | Method | Vizwiz | SQA | GQA | TextVQA | MME | POPE (F1) | Mean % |
|---|---|---|---|---|---|---|---|---|
| 2880 | Vanilla | 63.27 | 73.57 | 65.38 | 67.04 | 1901.01 | 86.18 | 100.00% |
| 640 | FastV | 59.96 | 68.27 | 61.85 | 60.90 | 1687.83 | 83.24 | 93.06% |
| | VisionZip | 60.57 | 71.89 | 63.04 | 63.88 | 1886.87 | 85.70 | 97.31% |
| | Ours | 61.12 | 72.29 | 63.94 | 63.42 | 1892.13 | 85.82 | **97.73%** |
| 320 | FastV | 56.74 | 66.48 | 55.11 | 44.56 | 1558.11 | 71.88 | 82.70% |
| | VisionZip | 58.82 | 70.55 | 60.84 | 59.57 | 1813.33 | 82.24 | 93.60% |
| | Ours | 59.67 | 71.44 | 62.08 | 60.68 | 1914.55 | 84.63 | **95.96%** |
| 160 | VisionZip | 56.63 | 69.66 | 57.86 | 53.99 | 1740.87 | 76.47 | 88.92% |
| | Ours | 58.50 | 71.19 | 60.30 | 56.61 | 1825.64 | 79.47 | **92.36%** |

## B  CLS ATTENTION SCORE DISTRIBUTION

Figure 3 and 4 illustrates an example image together with the attention score distribution of the `[CLS]` token across different regions in each layer of LLaVA-v1.5 visual encoder (CLIP-ViT-L/14-336). We observe that in the penultimate layer (layer 23), the `[CLS]` token's attention becomes highly concentrated on a small subset of tokens, whereas in the early layers the attention is more broadly dispersed. This pattern is not an isolated case but a consistent trend across inputs.

To further validate this observation, we sampled multiple images from TextVQA and visualized the `[CLS]` token attention distribution in the penultimate layer , as shown in Figure 4. Consistent with the earlier example, the attention scores exhibit strong concentration on a limited number of tokens. These results highlight the substantial redundancy present in visual tokens encoded by the VLM's visual backbone. More importantly, the pronounced concentration of `[CLS]` attention provides empirical support for our proposed global–local consolidation strategy, which leverages these attention patterns to merge redundant tokens into compact yet semantically informative representations.

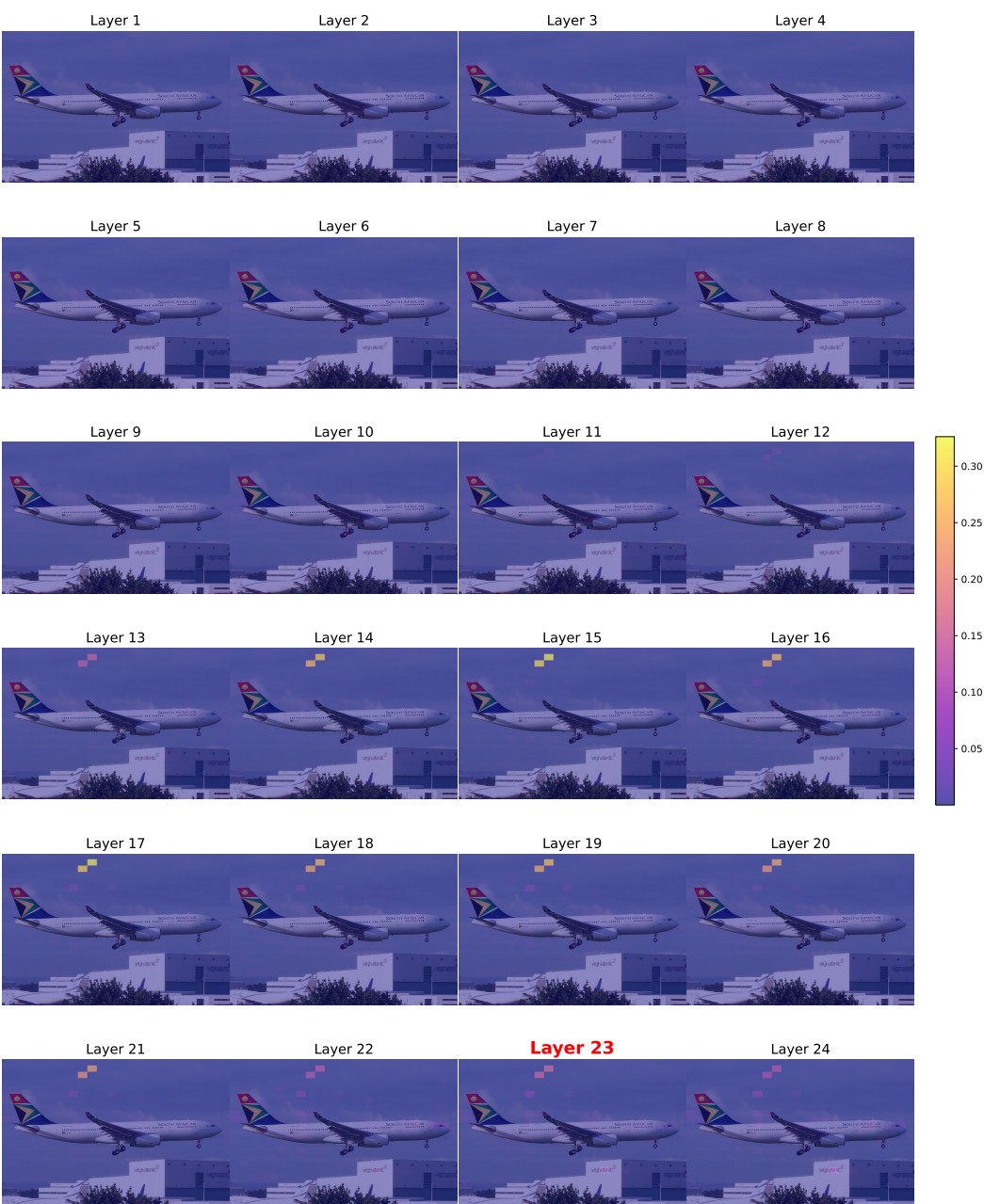

Figure 3: Attention score distribution of the `[CLS]` token across layers in the visual encoder of LLaVA-v1.5 (CLIP-ViT-L/14-336). In the early layers, attention is broadly dispersed, whereas in the later layers—particularly the penultimate one—it becomes highly concentrated on a small subset of tokens.

## C  THE USE OF LARGE LANGUAGE MODELS (LLMS)

We use large language models (LLMs) only to polish the writing, including grammar, clarity, and readability. The research ideas, technical framework, theoretical analyses, experimental design, and conclusions are entirely developed by the authors. The LLMs only improve the fluency and style of the manuscript and do not influence the originality, novelty, or scientific content of the work.

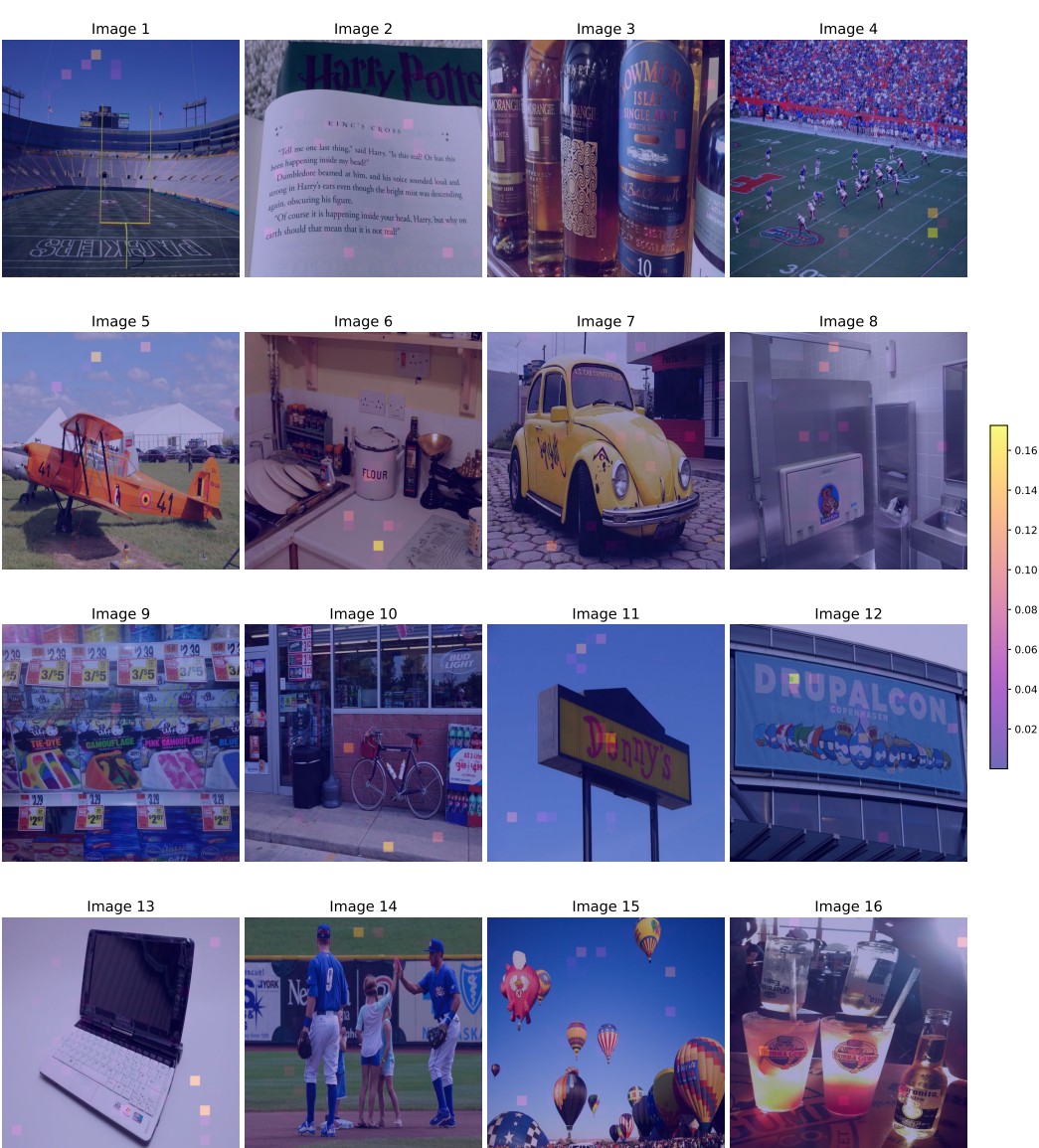

Figure 4: Visualization of the [CLS] token attention distribution in the penultimate layer across multiple images from TextVQA. The attention consistently concentrates on a limited number of visual tokens, revealing redundancy in the encoded representations.

