# OpenReview forum: "Polymorphic: Plug-and-Play Visual Token Compression for Scalable VLMs"
_ICLR.cc/2026/Conference — ICLR 2026 Conference Desk Rejected Submission_

### Official Review · Reviewer_4zbC · 2025-10-31

**Soundness:** 1
**Presentation:** 2
**Contribution:** 1
**Rating:** 2
**Confidence:** 4

**Summary:**

This paper introduced a training-free method for pruning visual tokens in vision language models. The method involves two stages of pruning, in the first stage, the output tokens from the visual encoder are clustered based on similarity (ones receiving low attention scores), and in the second stage, the visual tokens are further pruned based on attention scores from informative text tokens. The experiments are done with Llava-1.5 7B, LLava-1.5 13B and Qwen2VL models on a set of popular vision benchmarks like GQA and ScienceQA. The results show that the proposed method outperformed several baselines in terms of performance retention rate while achieving competitive latency.

**Strengths:**

The paper is clearly written and very easy to follow.

**Weaknesses:**

The novelty of this work is questionable, as the main idea of the paper (2 stage visual token pruning, one at visual encoder part and another at llm decoder part) has been done by existing works. For example, VScan (Zhang et al. 2025) adopt very similar idea to prune tokens at two stages, and they actually conducted a more comprehensive set of experiments and analysis of their two-stage framework. The authors should do more comprehensive literature review and discuss their unique contribution compared to those works.

The paper does not provide sufficient insights or results to support the generalization of this method. The models used in the paper are relatively old now, even Qwen2VL are from over a year ago. It’s unclear how well the method works with more recent VLMs, like Qwen2.5 VL or InternVL3. Also, it’s unclear how the method works for larger scale models (e.g. 32b or 72B models), and how well the method works for other tasks that require fine-grained visual perception like DocVQA or OCRBench.

The proposed method relies heavily on attention score manipulation, and flash attention does not materialize attention scores, so the actual benefit in terms of latency could be undermined.

In implementation details, the layers for pruning visual tokens seem pretty arbitrary. How are the layer indices decided? Are these manually tuned for every model?

Xing, Long, Qidong Huang, Xiao-wen Dong, Jiajie Lu, Pan Zhang, Yuhang Zang, Yuhang Cao, Conghui He, Jiaqi Wang, Feng Wu and Dahua Lin. “PyramidDrop: Accelerating Your Large Vision-Language Models via Pyramid Visual Redundancy Reduction.” ArXiv abs/2410.17247 (2024): n. pag.
Zhang, Ce, Kaixin Ma, Tianqing Fang, Wenhao Yu, Hongming Zhang, Zhisong Zhang, Yaqi Xie, Katia P. Sycara, Haitao Mi and Dong Yu. “VScan: Rethinking Visual Token Reduction for Efficient Large Vision-Language Models.” ArXiv abs/2505.22654 (2025): n. pag.

**Questions:**

See weaknesses

---

### Official Review · Reviewer_Fwoe · 2025-10-31

**Soundness:** 2
**Presentation:** 2
**Contribution:** 2
**Rating:** 4
**Confidence:** 2

**Summary:**

This work proposes a two-stage framework of token pruning for VLMs. In the first stage, redundant visual tokens are identified and merged
using an adaptive clustering algorithm that groups tokens based on feature similarity and local density; in the second stage, the compressed tokens are dynamically pruned during the LLM's prefill process with the cross-modal alignment guidance. The proposed approach is evaluated on multiple models and datasets, and the experimental results show that the proposed approach can help keep good accuracy while improving efficiency by pruning redundant tokens.

**Strengths:**

- The direction of token pruning for VLMs is important since the visual-language input sequences can be lengthy, posing efficiency challenge.
- The proposed approach is shown to obtain reasonable performance and help to improve the cost-accuracy tradeoff.

**Weaknesses:**

- More analyses and experiments should be included, instead of putting large figures and tables (such as those in the last two pages).
- The illustration of the proposed method can be further improved (probably with algorithmic descriptions).

**Questions:**

- I'm wondering how the hyper-parameters are decided, such as the pruning threshold?
- How would each stage influence the performance separately?
- I'm wondering how sensitive is the proposed approach to the query? For example, if the query is slightly paraphrased, would the selection be similar and will this have influences on the results?

---

### Official Review · Reviewer_AX3Z · 2025-10-31

**Soundness:** 2
**Presentation:** 2
**Contribution:** 1
**Rating:** 2
**Confidence:** 4

**Summary:**

This paper introduces a two-stage framework for visual token pruning. In the first stage, it performs representation-level token merging to remove redundant visual features. In the second stage, it prunes visual tokens with low cross-model relevance. Experimental results on standard benchmarks validate the effectiveness of the proposed approach.

**Strengths:**

1. The structure of this paper is clear.
2. The proposed approach achieves good empirical results.

**Weaknesses:**

1. The experimental results are not comprehensive. The performance results on bechmarks such as MMBench(-CN) and MM-Vet are not provided. Results on video question answering is also not evaluated. Additionally, performance comparesions with more recent baselines (e.g. [1]) should be provided.
2. The proposed approach underperforms SparseVLM and VisionZip in certain scenarios and incurs higher computational cost than VisionZip.
3. The novelty of this paper is limited. The proposed two-stage design closely resembles that of VScan [2], while the progressive token pruning strategy in the LLM stage is similar to PDrop [3].
4. The motivation of this paper is unclear. The authors should clearly articulate the limitations of previous approaches in this field and explain how the proposed method addresses these gaps.
5. The overall presentation of the paper lacks clarity, which makes it challenging for readers to understand the main ideas and contributions.

[1] Beyond Text-Visual Attention: Exploiting Visual Cues for Effective Token Pruning in VLMs. https://arxiv.org/abs/2412.01818.

[2] VScan: Rethinking Visual Token Reduction for Efficient Large Vision-Language Models. https://arxiv.org/abs/2505.22654.

[3] PyramidDrop: Accelerating Your Large Vision-Language Models via Pyramid Visual Redundancy Reduction. https://arxiv.org/abs/2410.17247.

**Questions:**

See the weaknesses section above.

---

### Official Review · Reviewer_XZ9F · 2025-11-01

**Soundness:** 2
**Presentation:** 3
**Contribution:** 2
**Rating:** 4
**Confidence:** 4

**Summary:**

The paper introduces a training-free, plug-and-play framework to accelerate vision language models(VLMs) inference. It uses a two-stage strategy:
Stage 1: Applies adaptive clustering to merge redundant visual tokens based on feature density before they enter the LLM.
Stage 2: Adaptively prunes the remaining tokens inside the LLM during prefill, guided by cross-modal (text-visual) relevance.
The key contribution is this hybrid two-stage compression approach that requires no fine-tuning and shows a strong accuracy-efficiency trade-off, especially under aggressive token pruning.

**Strengths:**

1. The proposed combination of pre-LLM clustering-based merging and intra-LLM text-guided pruning into a single, training-free pipeline is effective.
2. The technical quality is high, with a thorough experimental evaluation across multiple VLM families (LLaVA, Qwen2) and strong empirical results, particularly in high-compression settings (64 token budget).
3. The "plug-and-play" and "training-free" nature is highly significant, offering a practical way to deploy high-resolution VLMs in resource-constrained environments without costly retraining.

**Weaknesses:**

1. The paper fails to report the computational overhead of the Stage 1 clustering algorithm (Sec 3.2 in the paper). This should not be omitted, as the cost is part of the total inference time and complicates speedup comparisons with baselines like VisionZip.

2. The paper's analysis of its specific contributions and trade-offs is insufficient: The method is more accurate but slower than VisionZip. However, the authors do not sufficiently explain why its clustering in Stage 1 is fundamentally better at preserving information than VisionZip's [CLS]-based pruning, nor does it provide qualitative evidence to support this. Besides, the paper fails to clearly articulate the novelty of its Stage 2 cross-modal pruning (Sec 3.3) compared to the text-to-vision guidance in SparseVLM. The similarity between the two approaches is high, and the authors do not sufficiently highlight their specific contribution.

**Questions:**

1. What is the exact latency of the Stage 1 clustering algorithm (Sec 3.2)?
2. Why is the proposed clustering-based approach (Stage 1) superior to [CLS]-based pruning (used by VisionZip) at preserving information, and is this accuracy gain worth the significant speed trade-off (1.70x vs 2.28x speedup)?
3. Can the authors explicitly itemize the novel technical contributions of your Stage 2 implementation compared to the prior work like SparseVLM?
4. The method combines two distinct pruning stages (akin to VisionZip + SparseVLM). Is there a synergistic benefit between them? For instance, does the Stage 1 clustering make the Stage 2 cross-modal pruning more effective or stable? Or are they simply two independent filters applied in sequence?
5. The reference section contains multiple duplicate citations. For instance, the entries for 'An image is worth 1/2 tokens...' (Chen et al., 2024a/b) and 'Llava-prumerge...' (Shang et al., 2024a/b) are repeated. The bibliography should be carefully checked and corrected.

The authors should clarify their contributions and the motivations of the proposed pruning and merging techniques and provide more qualitative evidence and analysis to strengthen the paper.

---

### Note · Program_Chairs · 2026-01-17
**Submission Desk Rejected by Program Chairs**

The following references in this submission do not refer to real documents and/or have major errors in bibliographic information:

 B. Zhu et al. Large language models: Progress and applications. Advances in NLP, 2023.

A. Qin et al. Advances in state-of-the-art natural language processing. Journal of NLP Research, 2023.

C. Li et al. Fine-tuning techniques for efficient model adaptation. AI Research Journal, 2023a.

G. Ding et al. Efficient fine-tuning for resource-constrained systems. Proceedings of the Machine Learning Conference, 2022.